# Re-Evaluating Chemotherapy Dosing Strategies for Ovarian Cancer: Impact of Sarcopenia

Rushi Shah [1], Clarissa Polen-De [2], Michaela McGree [3], Angela Fought [3] and Amanika Kumar [4,*]

[1] Department of Obstetrics and Gynecology, Mayo Clinic, Rochester, MN 55905, USA; shah.rushiparasbhai@mayo.edu

[2] Department of Gynecologic Oncology, Summa Health, Akron, OH 44304, USA; dec@summahealth.org

[3] Department of Quantitative Health Sciences, Division Clinical Trials and Biostatistics, Mayo Clinic, Rochester, MN 55905, USA; mcgree.michaela@mayo.edu (M.M.); fought.angela@mayo.edu (A.F.)

[4] Department of Obstetrics and Gynecology, Division of Gynecological Surgery, Mayo Clinic, Rochester, MN 55905, USA

[*] Correspondence: kumar.amanika@mayo.edu

**Abstract:** We investigated the impact of sarcopenia on adjuvant chemotherapy dosing in advanced epithelial ovarian cancer (EOC). The chemotherapy dosing and toxicity of 173 eligible patients who underwent cytoreductive surgery and adjuvant chemotherapy at a single institution were analyzed. Patients with a skeletal muscle index less than $39 \text{ cm}^2/\text{m}^2$ measured on a CT scan were considered sarcopenic. Sarcopenic and non-sarcopenic patients were compared with regard to relative dose intensity (RDI), completion of scheduled chemotherapy, toxicity, and survival. A total of 62 (35.8%) women were sarcopenic. Sarcopenic women were less likely to complete at least six cycles of chemotherapy (83.9% vs. 95.5%, $p = 0.02$). The mean RDI for both carboplatin (80.4% vs. 89.4%, $p = 0.03$) and paclitaxel (91.9% vs. 104.1%, $p = 0.03$) was lower in sarcopenic patients compared to non-sarcopenic patients. Despite these differences in chemotherapy, there was no difference in neutropenia or median overall survival (3.99 vs. 4.57 years, $p = 0.62$) between the sarcopenic and non-sarcopenic women, respectively. This study highlights the importance of considering lean body mass instead of body weight or surface area in chemotherapy dosing formulas for sarcopenic women with advanced EOC. Further research is needed to optimize chemotherapy strategies based on individual body composition, potentially leading to improved dosing strategies in this population.

**Keywords:** sarcopenia; adjuvant chemotherapy; advanced epithelial ovarian cancer; relative dose intensity; cytoreductive surgery; overall survival

## 1. Introduction

Ovarian cancer is a highly aggressive and lethal gynecologic malignancy, contributing significantly to the global disease burden. In 2018 alone, an estimated 295,000 new cases and 185,000 deaths were attributed to ovarian cancer worldwide [1]. Among cancers affecting women, ovarian cancer ranks as the eighth most common cancer for both incidence and the cause of cancer-related deaths in females [1]. One of the key challenges in managing ovarian cancer is the lack of effective early diagnosis measures. This delayed diagnosis significantly contributes to the poor prognosis and high mortality rates associated with ovarian cancer [2]. Most patients are diagnosed at an advanced stage, requiring both high-complexity cytoreductive surgery and platinum-based chemotherapy. Optimizing treatment strategies in the advanced stage patient is crucial for ovarian cancer treatment.

Sarcopenia is a condition characterized by the gradual decline of muscle mass and strength, commonly observed in older individuals [3]. As people age, the prevalence of sarcopenia increases significantly. It affects approximately 5–13% of individuals aged 60 and above, and the prevalence increases to 11–50% in those aged 80 and above [4]. This musculoskeletal degeneration can hinder daily activities, leading to reduced mobility,

increased frailty, and an overall decline in functional capacity. Beyond its impact on physical function, sarcopenia has been recognized as a significant predictor of negative outcomes after surgery, particularly in major procedures like emergency abdominal surgery, hepatic resection, esophagectomy, and pancreatectomy. Patients with sarcopenia undergoing surgery are at higher risk of experiencing complications, increased mortality rates, and more postoperative morbidity [5–7].

Previous research has indicated that sarcopenia can serve as a prognostic factor linked to decreased survival rates and heightened resistance and toxicity to chemotherapy in patients diagnosed with different types of cancers, such as breast, small cell lung, urothelial, and gastric cancers [8–11]. In addition to the significant burden posed by ovarian cancer, emerging research has shed light on the potential impact of sarcopenia on patient outcomes in this malignancy. Sarcopenia has been associated with increased rates of chemotoxicity and reduced overall survival (OS) in ovarian cancer patients [12,13].

Several studies have explored the influence of sarcopenia on patient outcomes. These studies have repeatedly emphasized the importance of specific measurements as crucial indications, including the skeletal muscle index (SMI). These studies have specifically demonstrated that a lower baseline SMI or a fall in SMI over the course of treatment may negatively affect prognosis. However, the relationship between sarcopenia and patient outcomes in ovarian cancer remains complex and not fully understood. It is unclear whether SMI merely reflects the extent of disease, the burden of treatment, and compromised performance status, or if it serves as an independent prognostic factor [14,15].

The standard of care treatment for ovarian cancer includes the cytotoxic agents paclitaxel and carboplatin, both of which utilize some measure of body composition in dosing formulas. Paclitaxel dosing is based on body surface area (BSA). However, research on body composition has revealed that individuals with similar body weight, BSA, or body mass index (BMI) may have varying body compositions, especially when evaluating visceral fat and skeletal muscle [16]. Also, there is lack of substantial evidence of the correlation between physiologic functions such as hepatic and renal drug clearance and BSA [16]. Similarly, carboplatin dosing is determined based on renal function, specifically glomerular filtration rate (GFR), which is estimated by calculating creatinine clearance. Various formulas are employed in clinical practice for this calculation, including the widely used Cockcroft–Gault equation [17]. The equation incorporates weight as a factor, leading to debate about whether ideal or actual weight should be used [17]. There are concerns that using actual body weight may result in an overestimation of GFR and, subsequently, carboplatin dosing. The use of creatinine to estimate renal function is dependent on muscle mass. Specifically, sarcopenic patients with low muscle mass will have a lower creatinine and perhaps inappropriately elevated estimated renal clearance.

To date, few studies have examined the relationship between chemotherapy dosing, toxicity, and sarcopenia in advanced ovarian cancer. Thus, we aimed to investigate the relationship between sarcopenia and adjuvant chemotherapy delivery and toxicity among women with advanced EOC.

## 2. Materials and Methods

### 2.1. Patient Characteristics

This is a retrospective cohort study conducted at the Mayo Clinic, approved by the Institutional Review Board. The study focused on eligible women identified from the institution's prospectively maintained epithelial ovarian cancer (EOC) surgical database. Inclusion criteria comprised women who underwent primary cytoreductive surgery (PCS) for advanced stage (IIIC/IV) ovarian cancer (i.e., ovary, fallopian tube, or primary peritoneal origin) between 1 February 2003 and 31 December 2018. Additionally, participants needed to have received all their adjuvant chemotherapy at Mayo Clinic for inclusion in the study.

Exclusion criteria included patients with palliative-only surgery or any portion of chemotherapy at another institution, and those for whom sarcopenia data were unavailable. Furthermore, women who declined to grant access to their medical records for research pur-

poses were also excluded from the study. Data on patient characteristics, oncologic details, surgical outcomes [18], and intraoperative characteristics were abstracted. Chemotherapy data including type prescribed, amount (prescribed and actual doses), toxicity, recurrence, and vital status with date of last follow-up were also recorded.

To assess skeletal muscle and adipose tissue areas, a radiologist-selected axial CT image featuring fully visible transverse processes at the level of the third lumbar vertebrae was used for measurement of body composition using Slice-O-Matic software v4.3 (TomoVision) [19]. The software employed distinct attenuation thresholds (measured in Hounsfield units—HU) for differentiating skeletal muscle, subcutaneous and intramuscular adipose tissue, and visceral adipose tissue. Skeletal muscle area (SMA) was measured directly using cross-sectional CT by measuring and summing the area of core skeletal muscle groups, including the psoas, erector spinae, quadratus lumborum, transversus abdominis, external and internal obliques, and rectus abdominis muscles. The measured SMA was divided by height squared (measured in $m^2$) to calculate SMI [19]. Patients were then classified as sarcopenic if SMI was less than 39 $cm^2/m^2$, according to international sex-specific definitions [20,21].

We defined dose delay as a delay of 7 or more days in administering at least one chemotherapeutic agent during any chemotherapy cycle compared to the standard administration day [22,23]. A dose reduction was defined as at least one chemotherapeutic agent at a dose reduced by 15% or more during any chemotherapy cycle compared to the standard dose [22,23]. Moreover, in accordance with the medical records, dose reductions were established by taking into consideration multiple factors, such as adverse reactions, patient compliance, and clinical judgment by the attending physician. Severe neutropenia was defined as an absolute neutrophil count (ANC) of less than 1000 cells/$mm^3$, while febrile neutropenia was characterized by severe neutropenia accompanied by a fever greater than 38.3 degrees Celsius (101 degrees Fahrenheit).

The primary outcome of this study was to measure the relative dose intensity (RDI) specifically for patients with standard chemotherapy (intravenous carboplatin and paclitaxel for up to 18 weeks). Patients were also excluded if they did not have a planned regimen of carboplatin and paclitaxel. Patients were only included if we had all chemotherapy dosing information in the medical record. For the measurement of chemotherapy dosing, we used the standard definition to calculate BSA and GFR—the Mosteller formula and the Cockcroft–Gault equation, respectively—according to a prior publication [24]. In addition, we used the Calvert formula to calculate the carboplatin milligram dose needed to achieve a given AUC (area under the free carboplatin plasma concentration versus time curve) [25]. Standard doses were calculated based on individual baseline creatinine, height, and weight and assuming a standard of care dose of intravenous carboplatin AUC of 6 and intravenous paclitaxel 175 mg/$m^2$ every 3 weeks for a total of 6 cycles. The actual dose of chemotherapy was calculated by summing abstracting given doses from the chart. The RDI for each drug was calculated as the percentage of the standard dose that was administered using the following formulas:

$$\text{Carboplatin RDI \%} = (\text{Total administered dose of carboplatin}/\text{Total standard dose of carboplatin}) \times 100$$

$$\text{Paclitaxel RDI \%} = (\text{Total administered dose of paclitaxel}/\text{Total standard dose of paclitaxel}) \times 100$$

### 2.2. Statistical Analysis

We used a two-sample *t*-test for age and BMI and the chi-square test or Fisher's exact test for categorical variables to compare sarcopenic and non-sarcopenic patients. Progression-free survival (PFS) and overall survival (OS) following the date of the surgery were estimated using the Kaplan–Meier method. All calculated *p*-values are two-sided, and *p*-values < 0.05 were considered statistically significant.

### 3. Results

In this study, a total of 173 patients who met the inclusion criteria were included in the analysis. Among this cohort of women, 35.8% (62/173) of patients were categorized as sarcopenic based on the specified cut-off value for SMI being less than 39 cm$^2$/m$^2$. The patient, oncologic, and surgical characteristics of sarcopenic and non-sarcopenic women are listed in Table 1.

**Table 1.** Characteristics of non-sarcopenic versus sarcopenic women undergoing adjuvant chemotherapy after primary debulking surgery for stage IIIC-IV advanced epithelial ovarian cancer.

| Characteristic | Total $n = 173$ | No Sarcopenia $n = 111$ (64.2%) | Sarcopenia $n = 62$ (35.8%) | $p$ [†] |
|---|---|---|---|---|
| Age at chemotherapy (years), mean (SD) | 63.6 (10.5) | 61.4 (11.0) | 67.5 (8.2) | <0.01 |
| BMI at chemotherapy (kg/m$^2$), mean (SD) | 26.2 (5.8) | 28.0 (5.8) | 23.1 (4.0) | <0.01 |
| Skeletal muscle area (cm$^2$), mean (SD) | 108.3 (18.7) | 117.5 (15.4) | 91.6 (10.9) | <0.01 |
| Skeletal muscle index (cm$^2$/m$^2$), mean (SD) | 40.9 (7.3) | 44.8 (5.7) | 33.8 (3.5) | <0.01 |
| Skeletal muscle density (HU), mean (SD) | 33.8 (9.9) | 33.7 (10.3) | 34.1 (9.2) | 0.78 |
| ASA score, *n* (%) | | | | 0.64 |
| <3 | 102 (59.0) | 64 (57.7) | 38 (61.3) | |
| ≥3 | 71 (41.0) | 47 (42.3) | 24 (38.7) | |
| Preoperative albumin (g/dL), *n* (%) | | | | 0.40 |
| ≥3.5 | 133 (76.9) | 82 (73.9) | 51 (82.3) | |
| <3.5 | 21 (12.1) | 16 (14.4) | 5 (8.1) | |
| Not available | 19 (11.0) | 13 (11.7) | 6 (9.7) | |
| FIGO grade, *n* (%) | | | | 0.16 |
| 1–2 | 9 (5.2) | 8 (7.2) | 1 (1.6) | |
| 3 | 164 (94.8) | 103 (92.8) | 61 (98.4) | |
| FIGO stage, *n* (%) | | | | 0.32 |
| IIIC | 132 (76.3) | 82 (73.9) | 50 (80.6) | |
| IV | 41 (23.7) | 29 (26.1) | 12 (19.4) | |
| Histology, *n* (%) | | | | 0.99 |
| Non-serous | 13 (7.5) | 8 (7.2) | 5 (8.1) | |
| Serous | 160 (92.5) | 103 (92.8) | 57 (91.9) | |
| Surgical complexity, *n* (%) | | | | <0.01 |
| Low | 21 (12.1) | 14 (12.6) | 7 (11.3) | |
| Intermediate | 77 (44.5) | 61 (55.0) | 16 (25.8) | |
| High | 75 (43.4) | 36 (32.4) | 39 (62.9) | |
| Residual disease, *n* (%) | | | | 0.55 |
| Microscopic | 98 (56.6) | 65 (58.6) | 33 (53.2) | |
| Measurable (≤1 cm) | 64 (37.0) | 38 (34.2) | 26 (41.9) | |
| Suboptimal (>1 cm) | 11 (6.4) | 8 (7.2) | 3 (4.8) | |

Abbreviations: ASA, American Society of Anesthesiologists; BMI, body mass index; FIGO, International Federation of Gynecology and Obstetrics; SD, standard deviation. [†] Chi-square or Fisher's exact *p*-value reported for categorical variables and *t*-test reported for continuous variables.

According to Table 1, sarcopenic women were found to be older, with a mean age of 67.5 years, compared to 61.4 years in non-sarcopenic women ($p < 0.01$), but with no difference in functional status (ASA score). Furthermore, the study found that sarcopenic

women had a lower mean body mass index (BMI) of 23.1 kg/m$^2$, in contrast to 28.0 kg/m$^2$ in non-sarcopenic women ($p < 0.01$). Sarcopenic patients were more likely to have undergone surgeries with higher complexity when compared to non-sarcopenic patients [18]. Approximately 62.9% of sarcopenic patients had high surgical complexity, while only 32.4% of non-sarcopenic patients fell into this category ($p < 0.01$).

In this study, we calculated the RDI of carboplatin and paclitaxel for a total of 38 sarcopenic women and 78 non-sarcopenic women (as indicated in Table 2). RDI is a measure of the actual received dose of a chemotherapy drug compared to the standard dose. Upon analysis, we found that sarcopenic women received lower doses of chemotherapy compared to non-sarcopenic women. Specifically, the mean RDI for carboplatin in sarcopenic women was 80.4%, whereas it was 89.4% in non-sarcopenic women ($p = 0.03$). Similarly, the mean RDI for paclitaxel was 91.9% in sarcopenic women and 104.1% in non-sarcopenic women ($p = 0.03$). These findings indicate that sarcopenic women were more likely to receive reduced doses of both carboplatin and paclitaxel compared to their non-sarcopenic counterparts.

**Table 2.** Impact of sarcopenia on carboplatin and paclitaxel RDI.

| Characteristic | Total $n = 116$ | No Sarcopenia $n = 78$ | Sarcopenia $n = 38$ | $p$ [†] |
|---|---|---|---|---|
| Carboplatin RDI (%), mean (SD) | 86.5 (21.0) | 89.4 (18.8) | 80.4 (24.1) | 0.03 |
| Paclitaxel RDI (%), mean (SD) | 100.1 (27.9) | 104.1 (28.8) | 91.9 (24.4) | 0.03 |
| Carboplatin RDI $\geq$ 85%, $n$ (%) | 70 (60.3) | 50 (64.1) | 20 (52.6) | 0.24 |
| Paclitaxel RDI $\geq$ 85%, $n$ (%) | 93 (80.2) | 65 (83.3) | 28 (73.7) | 0.22 |

Abbreviations: BSA, body surface area; RDI, relative dose intensity, SD, standard deviation. [†] Chi-square reported for categorical variables and *t*-test reported for continuous variables.

However, we also assessed the RDI using a cut-off value of 85% to define an adequate dose. When using this 85% cut-off, the study did not find a statistically significant difference in RDI between sarcopenic and non-sarcopenic women for both carboplatin and paclitaxel. This suggests that, based on this specific cut-off, the difference in the actual dose received by sarcopenic and non-sarcopenic women did not reach a level of statistical significance, and both groups were considered to have received an adequate dose of chemotherapy.

According to the data presented in Table 3, we found a statistically significant difference between sarcopenic and non-sarcopenic women concerning the completion of at least six cycles of chemotherapy. Specifically, sarcopenic women were significantly less likely to complete the recommended six cycles of chemotherapy, with a completion rate of 83.9%, compared to 95.5% in non-sarcopenic women ($p = 0.02$). However, when examining other important treatment-related factors, we did not find significant differences between sarcopenic and non-sarcopenic women. There were no notable differences in the occurrence of dose delays, dose reductions, or severe neutropenia between the two groups, despite the lower likelihood of completing at least six cycles in the sarcopenic group. Regarding specific outcomes related to neutropenia, the data showed that sarcopenic women tended to have a lower incidence of febrile neutropenia (3.2% vs. 4.5%) and were less likely to receive Neupogen or Neulasta support (medications to stimulate white blood cell production) compared to non-sarcopenic women. However, it is essential to note that these differences did not reach statistical significance.

We evaluated patients' groups based on their carboplatin relative dose intensity (RDI) and the presence and absence of sarcopenia (Table 4). There is a statistically significant difference in age across these groups, with the "Carboplatin RDI < 85% and sarcopenia" group having the highest mean age (67.4 years), $p = 0.01$. Interestingly, the RDI < 85% group without sarcopenia did have the highest BMI, suggesting a role of body composition in chemotherapy dosing. Lastly, in terms of the distribution of patients based on the level of surgical complexity, the two non-sarcopenic groups were less likely to have high

surgical complexity, perhaps suggesting less overall disease burden and less need for complex surgeries.

Among 173 women, 93 deaths occurred and 126 experienced cancer progression within 5 years of surgery. The findings revealed that there was no statistically significant difference in OS between the sarcopenic and non-sarcopenic women (log-rank *p* = 0.62). The median OS for the sarcopenic group was 3.99 years, while it was 4.57 years for the non-sarcopenic group (Figure 1). Similarly, the study found no statistically significant difference in PFS between the sarcopenic and non-sarcopenic women (log-rank *p* = 0.86). The median PFS for the sarcopenic group was 1.46 years, while it was 1.48 years for the non-sarcopenic group (Figure 2).

**Table 3.** Chemotherapy toxicity for non-sarcopenic versus sarcopenic women undergoing adjuvant chemotherapy after primary debulking surgery for stage IIIC-IV advanced epithelial ovarian cancer.

| Characteristic | Total *n* = 173 | No Sarcopenia *n* = 111 | Sarcopenia *n* = 62 | *p* [†] |
|---|---|---|---|---|
| Febrile neutropenia, *n* (%) | | | | 0.99 |
| No | 166 (96.0) | 106 (95.5) | 60 (96.8) | |
| Yes | 7 (4.0) | 5 (4.5) | 2 (3.2) | |
| Severe neutropenia (grade 3–5), *n* (%) | | | | 0.27 |
| No | 88 (50.9) | 53 (47.7) | 35 (56.5) | |
| Yes | 85 (49.1) | 58 (52.3) | 27 (43.5) | |
| Dose delay, *n* (%) | | | | 0.35 |
| No | 92 (53.2) | 62 (55.9) | 30 (48.4) | |
| Yes | 81 (46.8) | 49 (44.1) | 32 (51.6) | |
| Dose reduction, *n* (%) | | | | 0.85 |
| No | 46 (26.6) | 29 (26.1) | 17 (27.4) | |
| Yes | 127 (73.4) | 82 (73.9) | 45 (72.6) | |
| Use of Neupogen or Neulasta during treatment, *n* (%) | | | | 0.31 |
| No | 126 (72.8) | 78 (70.3) | 48 (77.4) | |
| Yes | 47 (27.2) | 33 (29.7) | 14 (22.6) | |
| Completed at least six cycles, *n* (%) | | | | 0.02 |
| No | 15 (8.7) | 5 (4.5) | 10 (16.1) | |
| Yes | 158 (91.3) | 106 (95.5) | 52 (83.9) | |

[†] Chi-square or Fisher's exact *p*-value.

**Table 4.** Characteristics of patients undergoing carboplatin chemotherapy with different RDI and sarcopenia statuses.

| Characteristic | Carboplatin RDI < 85%, No Sarcopenia *n* = 28 | Carboplatin RDI < 85% and Sarcopenia *n* = 18 | Carboplatin RDI ≥ 85%, No Sarcopenia *n* = 50 | Carboplatin RDI ≥ 85%, Sarcopenia *n* = 20 | *p* [†] |
|---|---|---|---|---|---|
| Age (years), mean (SD) | 63.5 (11.0) | 67.4 (10.4) | 60.8 (10.6) | 68.4 (7.8) | 0.01 |
| BMI (kg/m$^2$), mean (SD) | 30.9 (6.9) | 23.6 (4.7) | 26.7 (4.6) | 22.4 (2.8) | <0.01 |
| ASA score, *n* (%) | | | | | 0.16 |
| <3 | 16 (57.1) | 10 (55.6) | 30 (60.0) | 17 (85.0) | |
| ≥3 | 12 (42.9) | 8 (44.4) | 20 (40.0) | 3 (15.0) | |

**Table 4.** *Cont.*

| Characteristic | Carboplatin RDI < 85%, No Sarcopenia $n = 28$ | Carboplatin RDI < 85% and Sarcopenia $n = 18$ | Carboplatin RDI $\geq$ 85%, No Sarcopenia $n = 50$ | Carboplatin RDI $\geq$ 85%, Sarcopenia $n = 20$ | $p$ [†] |
|---|---|---|---|---|---|
| Preoperative albumin (g/dL), *n* (%) | | | | | 0.68 |
| $\geq$3.5 | 20 (71.4) | 16 (88.9) | 38 (76.0) | 17 (85.0) | |
| <3.5 | 6 (21.4) | 1 (5.6) | 6 (12.0) | 1 (5.0) | |
| Not available | 2 (7.1) | 1 (5.6) | 6 (12.0) | 2 (10.0) | |
| Surgical complexity, *n* (%) | | | | | 0.02 |
| Low | 3 (10.7) | 2 (11.1) | 7 (14.0) | 2 (10.0) | |
| Intermediate | 18 (64.3) | 5 (27.8) | 28 (56.0) | 5 (25.0) | |
| High | 7 (25.0) | 11 (61.1) | 15 (30.0) | 13 (65.0) | |

Abbreviations: ASA, American Society of Anesthesiologists; BMI, body mass index; SD, standard deviation. [†] Chi-square or Fisher's exact *p*-value reported for categorical variables and Kruskal-Wallis test *p*-value reported for continuous variables.

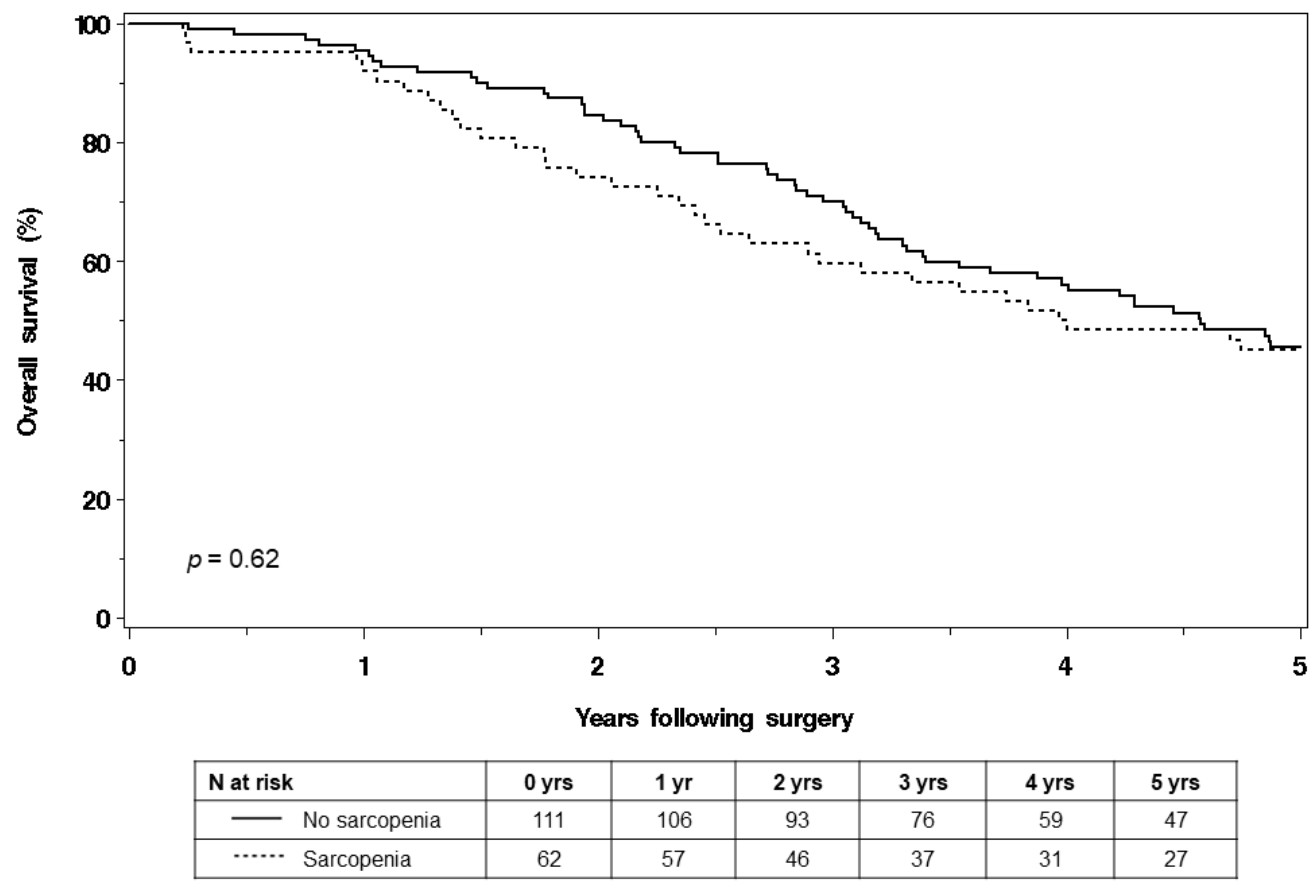

| N at risk | 0 yrs | 1 yr | 2 yrs | 3 yrs | 4 yrs | 5 yrs |
|---|---|---|---|---|---|---|
| —— No sarcopenia | 111 | 106 | 93 | 76 | 59 | 47 |
| ······ Sarcopenia | 62 | 57 | 46 | 37 | 31 | 27 |

**Figure 1.** Overall survival of sarcopenic and non-sarcopenic women.

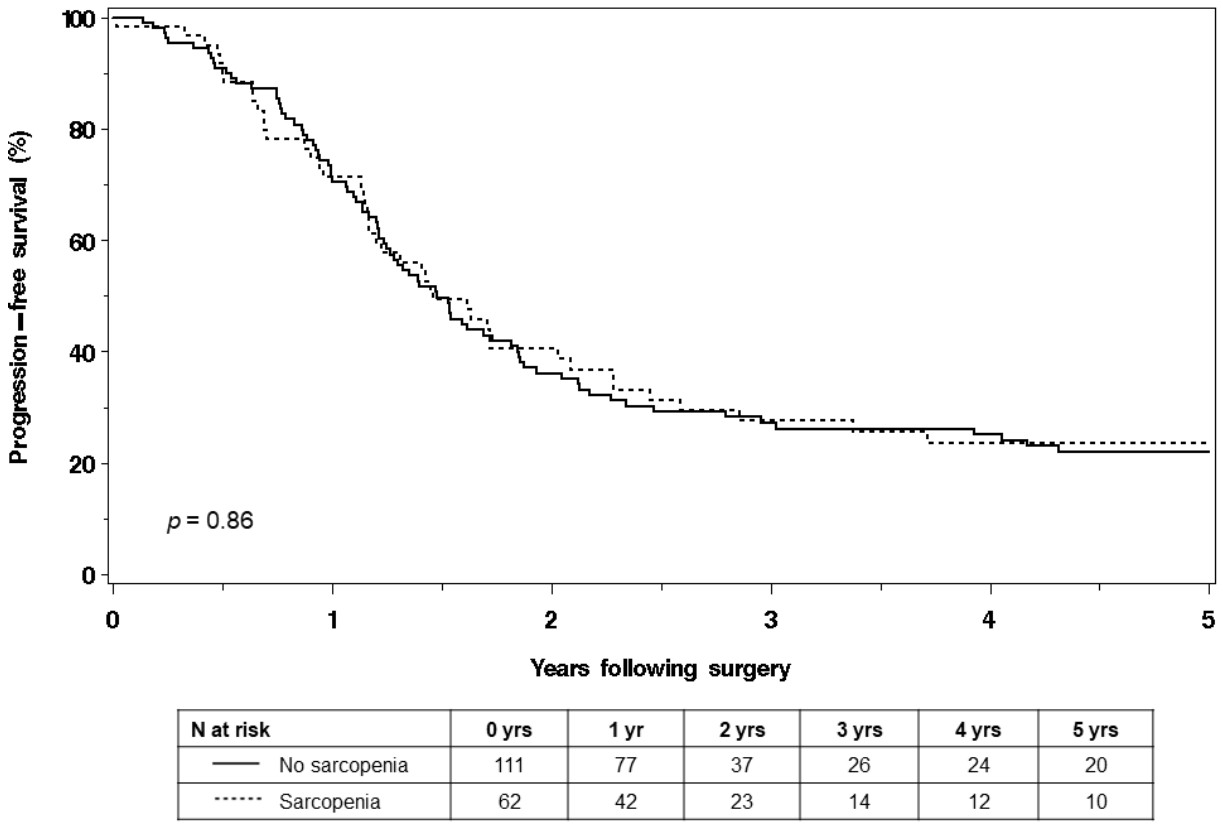

| N at risk | 0 yrs | 1 yr | 2 yrs | 3 yrs | 4 yrs | 5 yrs |
|---|---|---|---|---|---|---|
| ——— No sarcopenia | 111 | 77 | 37 | 26 | 24 | 20 |
| ······ Sarcopenia | 62 | 42 | 23 | 14 | 12 | 10 |

**Figure 2.** Progression-free survival of sarcopenic and non-sarcopenic women.

### 4. Discussion

This study examines the relationship between body composition (specifically sarcopenia) and adjuvant chemotherapy in women with advanced EOC. Overall, sarcopenic women were older and had a lower BMI compared to the non-sarcopenic women. Sarcopenic women were significantly less likely to complete the standard of care of six cycles of chemotherapy, with a lower overall RDI for both carboplatin and paclitaxel. They also were more likely to have high-complexity surgery, suggesting a risk for more complications and more disease burden. However, those patients with an RDI < 85% were not significantly older and did not have more complexity. Sarcopenic women do not have a higher frequency of dose delay, dose reductions, or neutropenia compared to non-sarcopenic women and are equally likely to achieve $\geq$ 85% RDI.

Several studies have looked at the prevalence of sarcopenia in cancer patients. In a comprehensive meta-analysis conducted by Ubachs et al. (2019), the prevalence of sarcopenia in ovarian cancer patients was found to range widely, from 11% to 54%. This considerable variation in rates could be attributed to the utilization of different cut-off values for diagnosing sarcopenia, specifically ranging from 38.5 to 41.5 cm$^2$/m$^2$ [26]. Rutten et al. [13] conducted a study involving 109 patients with advanced ovarian cancer and reported a sarcopenia prevalence of 55%. The higher incidence of sarcopenia in this study could be attributed to the use of a specific cut-off value for SMI, which was set at 41.5 cm$^2$/m$^2$. This choice of cut-off appears to have contributed to the comparatively higher rate of sarcopenia observed in their investigation [13]. In our study, we observed a sarcopenia rate of 35.8% when employing an SMI cut-off value of 39 cm$^2$/m$^2$. This finding aligns with previous studies, demonstrating consistency in the prevalence of sarcopenia among the investigated population.

The connection between reduced lean mass and heightened vulnerability to chemotherapy toxicity has been demonstrated in both early- and late-stage disease, regardless of the specific type of cancer or systemic chemotherapy employed [27–29]. While most studies

have observed a correlation between diminished lean mass and poorer treatment tolerance, a few smaller studies have reported no such association [30–32]. The increased toxicity observed in patients with low lean mass can be attributed to potential alterations in the distribution, metabolism, and clearance of systemic chemotherapy drugs [33]. Body weight consists of two primary components: lean mass and fat mass. These two components serve as major sites for the distribution of hydrophilic and lipophilic drugs [34,35]. So, variations in individual lean or fat mass can lead to changes in the drug's volume of distribution, consequently affecting the tolerance to cytotoxic drugs [36]. Supporting this hypothesis, pharmacokinetic data have indicated that patients with low lean mass may experience higher plasma concentrations of antineoplastic drugs, resulting in an increased likelihood of toxicity [37,38]. Additionally, it is worth noting that these patients exhibit excessive frailty and heightened susceptibility to acute medical events, which can further worsen the toxicity associated with chemotherapy [39].

Our study did not find a significant link between increased toxicity and sarcopenic women. However, it is important to note that our findings were based on a small sample size of patients, and we only focused on neutropenia and neutropenic fever, given the retrospective nature of the study. We did not consider other potential toxic effects of carboplatin and paclitaxel, such as fatigue, neuropathy, vomiting, and myalgias due to their lack of standardized collection in the medical record, which could contribute to a more comprehensive understanding of its toxicity spectrum [40]. We hypothesize that these toxicities differ between patients and believe the prospective collection of these data would be helpful in informing dosing alterations in sarcopenic women. It is essential to consider these limitations when interpreting our findings and to conduct further research to investigate the broader range of toxicities associated with these medications.

Sarcopenic women had a lower mean RDI of carboplatin and paclitaxel compared to the non-sarcopenic women as sarcopenic patients are more likely to have chemotoxicity than non-sarcopenic patients [27–29]. Furthermore, sarcopenic patients are older than the non-sarcopenic patients, which results in decreased performance status. We have previously shown a relationship between frailty and lower RDI [24]. One retrospective analysis conducted by R.K. Hanna et al. examined 325 women with advanced-stage ovarian cancer and found that maintaining a higher relative dose intensity of chemotherapy was associated with improved OS, although not with PFS [41]. Another study by Repetto et al., involving 226 women treated in randomized clinical trials with platinum-based chemotherapy regimens for stage III–IV ovarian cancer, did not find any significant correlation between RDI and response rates, PFS, or OS [42]. In another analysis by Fauci et al., which included 138 women with epithelial ovarian cancer of any stage treated with intravenous carboplatin and paclitaxel, it was observed that patients with an RDI between 70% and 110% had better PFS compared to those outside this range [43].

In our study, we found that there was no statistically significant difference in OS and PFS between the sarcopenic and non-sarcopenic women. The impact of sarcopenia on survival outcomes in ovarian cancer remains inconclusive, unlike other types of cancer where sarcopenia is generally associated with reduced OS and increased postoperative complications [44,45]. A few retrospective studies, conducted by Bronger et al., Kumar et.al, Polen-De et.al., and Rutten et al., exemplify the conflicting findings in this area. Bronger et al. focused on advanced-stage serous ovarian cancer and found that baseline sarcopenia independently predicted poorer prognosis in terms of PFS and OS [14]. In other words, patients with sarcopenia at the beginning of their treatment experienced worse outcomes in terms of disease progression and OS compared to those without sarcopenia [14]. This study differed from our current study as it did not account for the receipt of adjuvant chemotherapy or a center for chemotherapy, likely reflecting a different population with regard to surgical outcomes.

In addition, Kumar et al. have shown that, among women with advanced ovarian cancer undergoing primary cytoreductive surgery, the presence of sarcopenia was linked to a decline in OS, while no significant impact on PFS was observed [46]. However, Polen-

De et al. have shown a strong association between preoperative sarcopenia and adverse survival outcomes in older patients diagnosed with advanced ovarian cancer [47]. This study's results showed that older patients exhibiting sarcopenia before undergoing surgery faced significantly elevated risks of poor survival and increased mortality rates [47]. In contrast, Rutten et al. investigated ovarian cancer patients undergoing primary debulking surgery (PDS) and found that sarcopenia did not have a significant impact on OS or major complications [15]. This suggests that sarcopenia did not strongly influence OS or the occurrence of major complications in the context of PDS for ovarian cancer.

The strengths of this study include its commitment to an objective assessment of sarcopenia, which is rooted in the meticulous analysis of radiological data extracted from patients' medical records. It is important to underscore that our study deliberately focused on a specific demographic, namely women with advanced epithelial ovarian cancer (EOC) classified as stage IIIC/IV, who were concurrently undergoing adjuvant chemotherapy. By exclusively including this subset of patients, we sought to create a more homogenous study cohort, thereby enhancing the internal validity of the results. All patients under investigation were treated at a single tertiary care center. This uniform care setting is crucial as it ensures that the treatment and management of advanced EOC patients are standardized across the study cohort. This approach minimizes the potential confounding variables that could arise from differing treatment protocols or methodologies across multiple medical institutions.

There are several limitations worth noting in the current study. Firstly, the sample size was small, and the retrospective study design may have introduced selection bias. This could potentially impact the generalizability of the findings. Secondly, the study did not account for the sequential changes in body composition within each individual over time, which could have provided more comprehensive insights. Thirdly, due to limitations in the available medical records, the evaluation of non-hematological complications such as neuropathy, myalgia, and fatigue was not possible. This lack of information may have influenced the decision-making process and the overall understanding of the outcomes. Lastly, it is important to consider that the cohort of patients in this study underwent primary surgery followed by adjuvant chemotherapy. Therefore, the generalizability of these results to sarcopenic women who receive neoadjuvant chemotherapy remains unclear and requires further investigation.

## 5. Conclusions

The main goal of this study was to investigate thoroughly how sarcopenia affects the results of adjuvant chemotherapy in patients with advanced ovarian cancer. One of the principal findings of this study is that sarcopenic women were less likely to complete six cycles of chemotherapy. Additionally, sarcopenic woman showed overall lower overall relative dose intensity for carboplatin and paclitaxel. However, there was no significant difference in dose delays, dose reductions, or neutropenia. The study highlights the importance of considering lean body mass in chemotherapy dosing for sarcopenic women. Further research is needed to optimize chemotherapy strategies based on individual body composition, potentially leading to improved dosing strategies in this population.

**Author Contributions:** Conceptualization, R.S. and A.K.; methodology A.K., R.S., M.M. and A.F.; formal analysis, A.F. and M.M.; investigation, R.S. and A.K.; resources, A.K. data curation, R.S.; writing—original draft preparation, A.K. and R.S.; supervision, A.K.; Writing—Review and Editing, A.K., R.S., M.M., A.F. and C.P.-D. All authors have read and agreed to the published version of the manuscript.

**Funding:** This research received no external funding.

**Institutional Review Board Statement:** The study was conducted in accordance with the Declaration of Helsinki and approved by the Institutional Review Board (or Ethics Committee) of Mayo Clinic (13-002493 01/08/2023 for studies involving humans.

**Informed Consent Statement:** Patient consent was waived due retrospective nature of the study.

**Data Availability Statement:** De-identified data available upon request to the corresponding author.

**Conflicts of Interest:** The authors declare no conflict of interest.

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
