# Peer review of "Re-Evaluating Chemotherapy Dosing Strategies for Ovarian Cancer: Impact of Sarcopenia"

_curroncol, doi:10.3390/curroncol30110688_

Round 1

Reviewer 1 Report

Comments and Suggestions for Authors

Overall it is a well written manuscript that I found interesting to read. It is a retrospective study and therefore it has limited overall merit but it does ask important questions 

I would like to see an explanation why only some of the women were analysed for RDI, (line 178) , it is not clear from the text. 

Author Response

We have provided with the information why only some women were analyzed for the RDI in  the methodology section of the research paper (line 132). in summary, we have exculded women who had chemotherapy >18 weks, did not have planned chemotherapy of carboplatin and paclitaxel and medical records did not have all the dosing information of chemotherapy.

Reviewer 2 Report

Comments and Suggestions for Authors

Thanks for inviting me to review this manuscript. This topic is interesting. Is it reasonable to use BSA to caculate drug dosage for those patients who are extremely thin or fat? This is a common clinical problem. The author design this retrospective study to answer this question. The whole paper is written well and easy to read. The objective is clear, and the research method is well designed.

In results, there two questions needed to be clarified. (1) What is the main reason for  RDI<85% in sarcopenia group, and is there any difference between two groups? (2) Over 70% of patients in both groups have dose reduction (Table 3). What is the reason for reduction? clinical routine or physician desicion?

Author Response

Thank you for your thoughtful response.

  1. There could be several reasons for the sarcopenia to have mean RDI<85%. Sarcopenic patients are more likely to have chemothoxicity than non -sarcopenic patients. Also, Sarcopenic patients are older than the non-sarcopenic patinets which results in decreased performance status.  In this type of hypothesis-generating retrospective research, I don't think we are able to answer that but will be the basis of future research
  2. according to the medical records, the dose reduction were determined on the basis of the various factors such as adverse effects, patient compliance and physician decision.  We can't reliable determine this based on the medical record review and so have not included it

Reviewer 3 Report

Comments and Suggestions for Authors

In this article "Re-evaluating chemotherapy dosing strategies for ovarian cancer: impact of sarcopenia" Rushi Shah et al investigate how sarcopenia may affect adjuvant chemotherapy treatment in patients with advanced ovarian cancer.

The abstract is concrete and explains the work well. The English used is clear and shows no flaws. The work is well organized and presented.

Regarding the methodology, I do have some doubts.

The authors compare two groups that are not homogeneous. There are statistically significant differences in the age of the patients, in their BMI and in the surgical complexity. Given that the intention is to study the results of adjuvant treatment in these patients, these factors could be very important biases.

I recommend performing a propensity score to match the two groups and to be able to compare truly homogeneous groups with the exception of sarcopenia status.

At the present stage, it is difficult to draw conclusions from the comparison of clearly different groups.

Author Response

Thank you.  You point is very good however I believe seeing that the groups are quite different is the very point we wanted to make.  We do need in medical oncology better objective ways of personalized chemotherapy prescription, and sarcopenia may be one of those easily obtainable, personalized options for futurue considieration.

Propensity score will have many biases and limitations, and actually instead of trying to balance groups, can lead to worsening bias and more imbalance and model dependence.  There is a large literature on this.  (one example, https://doi.org/10.1378/chest.12-1920).  So we believe by showing these more general data that include body composition and show that low muscle mass is 1) common in ovarian cancer and 2) correlated with lower chemotherapy dosing that we will be able to more accurately generate sound future hypotheses for future research. 

Round 2

Reviewer 3 Report

Comments and Suggestions for Authors

Thanks for yours answers

Understanding that the groups are different, I still find it remarkable that there is a difference in the age of the patients and in the surgical complexity. If it were possible, I would like to reflect a little more on whether this could have influenced compliance or not with subsequent treatment. It is not developed if the increase in surgical complexity resulted in a greater number of postoperative complications that could limit chemotherapy.

Author Response

We have added a table that looks at age, BMI, surgical complexity between sarcopenic and non sarcopenic groups and chemotherapy RDI in response.  While sarcopenic patients were older and more likely to have high complexity surgery, those with an RDI< 85% were not necessarily older or have more complex surgery.